# Synthesis and preclinical application of a Prussian blue-based dual fluorescent and magnetic contrast agent (CA)

Nikolett Hegedűs[1], László Forgách[1]*, Bálint Kiss[1‡], Zoltán Varga[1,2‡], Bálint Jezsó[2‡], Ildikó Horváth[1], Noémi Kovács[1‡], Polett Hajdrik[1‡], Parasuraman Padmanabhan[3,4], Balázs Gulyás[3,4,5], Krisztián Szigeti[1], Domokos Máthé[1,6,7]

1 Department of Biophysics and Radiation Biology, Semmelweis University, Budapest, Hungary, 2 Institute of Materials and Environmental Chemistry, Research Centre for Natural Sciences, Hungarian Academy of Science, Budapest, Hungary, 3 Lee Kong Chian School of Medicine, Nanyang Technological University, Singapore, Singapore, 4 Cognitive Neuroimaging Centre, Nanyang Technological University, Singapore, Singapore, 5 Department of Clinical Neuroscience, Karolinska Institute, Stockholm, Sweden, 6 CROmed Translational Research Centers, Budapest, Hungary, 7 In Vivo Imaging Advanced Core Facility, Hungarian Center of Excellence for Molecular Medicine (HCEMM), Budapest, Hungary

☯ These authors contributed equally to this work.
‡ BK, ZV, BJ, NK and PH also contributed equally to this work.
* forgach.laszlo@med.semmelweis-univ.hu

**Data Availability Statement:** Yes - all data are available without restriction. Supplemental information to the results presented in the study

## Abstract

The aim of this study was to develop and characterize a Prussian Blue based biocompatible and chemically stable T1 magnetic resonance imaging (MRI) contrast agent with near infrared (NIR) optical contrast for preclinical application. The physical properties of the Prussian blue nanoparticles (PBNPs) (iron (II); iron (III);octadecacyanide) were characterized with dynamic light scattering (DLS), zeta potential measurement, atomic force microscopy (AFM), and transmission electron microscopy (TEM). In vitro contrast enhancement properties of PBNPs were determined by MRI. In vivo T1-weighted contrast of the prepared PBNPs was investigated by MRI and optical imaging modality after intravenous administration into NMRI-Foxn1 nu/nu mice. The biodistribution studies showed the presence of PBNPs predominantly in the cardiovascular system. Briefly, in this paper we show a novel approach for the synthesis of PBNPs with enhanced iron content for T1 MRI contrast. This newly synthetized PBNP platform could lead to a new diagnostic agent, replacing the currently used Gadolinium based substances.

## Introduction

Magnetic resonance imaging (MRI) is widely used in the clinics as the pre-eminent whole body diagnostic tool to resolve morphology and functionality of the human body. Oftentimes the intrinsic MR contrasts of different biological tissues are not effective enough for high spatial resolution imaging. In these cases, the application of extrinsic contrast agents (CA) is mandatory. These materials are responsible for the shortening of transverse T2/T2* (negative

are available from figshar at: https://doi.org/10.6084/m9.figshare.17260385.v1.

**Funding:** This study was funded by H2020 European Institute of Innovation and Technology in the form of a grant [739593], by the Semmelweis University in the form of funds to NH [STIA-KFI-2020], and by the János Bolyai Research Fellowship Program of the Hungarian Academy of Science in the form of funds to ZV and KS. This study was also funded in part by the Thematic Excellence Programme (TKP) of the Ministry of Innovation and Technology of Hungary, within the framework of the BIOImaging Excellence programme at Semmelweis University, in the form of funds to LF.

**Competing interests:** The authors have declared that no competing interests exist.

contrast), or the longitudinal T1 (positive contrast) relaxation times of water molecules, and their application leads to negative (dark contrast) or positive (enhanced light contrast) areas in images [1, 2].

Iron-based particles, as MR active CAs [3, 4] have been studied for well in the past [5–7]. Their known T2/T2* shortening properties are translated into very large values for r2 (transverse relaxivity). This produces a negative contrast in MR images. Because of this, their clinical application is difficult as it is not always possible to differentiate if the signals come from the CA or a different biological tissue which is rich in blood, calcium, or other metals. It is also not convenient to medically report and monitor darkened areas in MR image instead of bright spots. Altogether, these factors contributed to the limited clinical application of iron based CAs and the intense research for positive T1 MR contrast materials [8–10].

Gadolinium (III)-containing CAs, as positive MR contrast materials seemed ideal T1 CA in the clinics during the 2000s [11, 12]. They cause hyperintense regions on the MR scans with excellent temporal and spatial resolution [13]. However, the European Commission has withdrawn all linear chelator-bound Gd-based contrast agents from the market in June 2018 based on the declaration of the European Medicines Agency (EMA; EMA/625317/2017) dated 19 December 2017. That declaration described those types of contrast agent might lead to severe toxic effects due to their accumulation in the brain. Further application of nonlinear, macrocyclic Gd contrast agents could be questioned as well due to their recently published adverse effects. It also can be assumed that the marketing authorizations of most of the presently applied Gd contrast agents could be cancelled.

In recent years, iron-based nanoparticle based T1 CAs became one of the most intensively researched domains in radiology [14]. They are built up from a magnetic core–it has size-dependent MR contrast property–and a biocompatible coat that reduces surface tension and ensures the colloidal stability of the sample. Their variable ultra-small size (1–500 nm), superparamagnetic behavior, biocompatibility, and chemical stability position them among the most frequently studied nanomaterials for biomedical applications. The final properties of their formulations depend on the combination of magnetic core and the coating. [2].

The blue nano-sized iron-based Prussian Blue precipitate (Prussian blue nanoparticles, PBNPs) seems to be an ideal CA base for T1 magnetic imaging. It has been used for nearly 300 years in electrochemical and biochemical experiments [15, 16]. In 2003 Prussian blue nanoparticle has been authorized and released (Radiogardase®) by the Food and Drug Administration (FDA) for human use. Originally, this compound was used for the treatment of heavy metal poisoning relying on the complexing property of Prussian blue [17]. Several methods are known for the synthesis (e.g., direct, or indirect) of Prussian blue nanoparticles with different shapes, sizes and stability depending on the applied method [18–20]. Due to the associated biocompatible shell comprising of organic acids and polymers, nanoparticles can be hidden from the immune cells and their biological half-life can be increased in the circulation [21, 22].

One of the widely used capping agents is citric acid, which directly affects the particle size. The higher the citric acid concentration, the smaller the particle size due to an increased reduction rate of the solution. This finding suggests that pH plays a crucial role during the synthesis of PBNPs [23].

Native Prussian blue nanoparticles show very weak contrast in T1 and T2-enhanced in vitro MRI images. Their measured relaxation times did not allow their application for in vivo studies. But it is known from the literature that both the size and the content of the particles and the ordered structure of nearby water molecules have significant effects on the MR contrast-enhancing properties of nanoparticles [24].

In pre-clinical routine, anatomical MR scans are often coupled with some higher functional contrast providing imaging modalities. Optical imaging (OI) could be an appropriate modality

associated with the MR technique for diagnostic and molecular imaging purposes due to their safety, relatively low cost as well as the high spatial resolution and real-time imaging capability. The only minor disadvantage of this technique is the limit of penetration depth of the applied light due to its scattering and absorption. However, this weakness is less of an issue by intraoperative guidance when tumors are directly revealed by the surgeon [25]. Furthermore, the application of near infrared light (NIR; 650–900 nm) for OI excitation could increase the light penetration into tissue up to 0.5–1.5 mm due to the weakened absorption of tissue chromophores, including oxyhemoglobin, deoxyhemoglobin, and melanin, while the scattering of the applied excitation light is negligible [26, 27]. Based on these features, OI is on its way to become a widely adopted method for tumor detection and image-guided surgery in the clinics [28–30].

Furthermore, with theragnostic outlook, PBNPs with their strong optical absorbance in the above-mentioned NIR window and excellent thermal conversion capability have been considered [31] as efficient photoacoustic contrast agents and they could act as an ideal imaging agent [17, 32–35].

For these reasons, our aim was to highly improve the T1 MR signal of Prussian blue particles with appropriate particle sizing and coating, while we also wished to eliminate the T2 contrast from the system. Additionally, the conjugated IR820 NIR fluorescent dye ensures higher tissue contrast in superficial regions even at lower concentration of dye. We aimed at a synthesis method that leads to a stable PB-CA for dual in vivo imaging.

## Materials and methods

### Citrate coated PB production

Citrate-coated PBNPs were produced with the process as described by Shokouhimehr [36]. A two-step PBNP preparation was made. Reactant solutions were made first, Solution A containing 20 mL of 1.0 mM Fe (III) chloride anhydrous (FeCl3; Merck KGaA, Darmstadt, Germany) with 0.5 mmol of citric acid (Merck KGaA), while Solution B contained 20 mL of 1.0 mM anhydrous potassium ferrocyanide (K4[Fe(CN)6]; Merck KGaA, Darmstadt, Germany) with 0.5 mmol citric acid (Merck KGaA) solution. Next, these solutions were mixed using fast stirring for 10 min at 60˚C.

### Production of uncoated PBNPs

Native PBNPs were synthesized again according to as described by Shokouhimehr [36], with modifications. As first step, the reactant solutions were made with Solution A contains 20 mL of 1.0 mM Fe(III) chloride anhydrous (FeCl3; Merck KGaA, Darmstadt, Germany) with 6 drops of 1 N HCl (Merck KGaA), while Solution B contained 20 mL of 1.0 mM potassium ferrocyanide anhydrous (K4[Fe(CN)6]; Merck KGaA, Darmstadt, Germany) with 6 drops of 1 N HCl (Merck KGaA). Secondly, these solutions were mixed slowly under vigorous stirring for 10 min at 60˚C.

### Preparation of fluorescent PBNP nanoparticle complexes

Following the coated and uncoated particle syntheses, the two different types of PBNPs were mixed under vigorous stirring for 10 min at 60˚C. With 10 minutes passed, 5 g Chelex 100 (chelating ion exchange resin, Merck KGaA, Darmstadt, Germany)/100 mL solution was applied to eliminate the superfluous metal or alkali metal ions from the system [37]. This suspension was stirred and incubated for one hour, whereby the styrene divinylbenzene copolymer beads were separated from the PBNP solution. In the next step, PBNPs were isolated from the complex suspension using ultracentrifugation (Eppendorf 5424R centrifuge, 21130 rcf) at 4˚C for 30 min.

We also produced a batch of uncoated PBNPs and fluorescent PBNP complexes, using the same method with a slight modification. Subsequently the production of reaction solutions of uncoated and complex PBNPs, we included an additional step of differential velocity centrifugation in the synthesis. The reaction of uncoated PBNP solutions were sedimented (Eppendorf 5424R centrifuge) 2 at times 1000 rcf and 2 times at 2000 rcf for 10 minutes consecutively. The PBNP complexes were centrifuged (Eppendorf 5424R centrifuge) 2 times at 2000 rcf for 10 minutes; one batch was filtered through a 0.22 μm membrane filter (MILLEX GP 0.22 μm; Merck KGaA, Darmstadt, Germany) and centrifuged at 2000 rcf for 10 minutes. As a final step, we isolated the particles by ultracentrifugation (Eppendorf 5424R centrifuge, 21130 rcf) at 4˚C for 30 minutes. To achieve fluorescence in the PBNPs, 0.1 mg/mL IR820 NIR dye was filtered through a 0.22 μm membrane filter (MILLEX GP 0.22 μm; Merck KGaA, Darmstadt, Germany). 10 μL of this filtered dye solution was adsorbed to the particles in 300 μL PBNP solution for a one-hour incubation.

### Dynamic light scattering (DLS) and Zeta measurement

The surface charge and hydrodynamic diameter of the particles were determined using a Litesizer 500 (Anton Paar, Hamburg, Germany). DLS measurement was performed at 25˚C in automatic mode (for backscatter detector fixed at 175˚; for side scatter 90˚ detector angle; for front scatter 15˚ detector angle) using a 633 nm He-Ne laser. Samples were measured in Omega cuvettes (Anton Paar, Hamburg, Germany). Measurement of zeta potential was performed under similar conditions. The measurement data were evaluated using software provided by the manufacturer, and statistical data and graphs were created and evaluated with Origin 9.0 (OriginLab) and Microsoft Excel 2013 software. DLS measurements were performed weekly for a period of 4 weeks to determine colloidal stability. Samples were stored at 4˚C.

### Transmission Electron Microscopy (TEM)

Morphological investigations of the NPs were carried out on a JEOL TEM 1011 TEM (JEOL, Peabody, MA, USA) operated at 80 kV. The camera used for image acquisition was a Morada TEM 11 MPixel from Olympus (Olympus, Tokyo, Japan) using iTEM5.1 software for metadata analysis. Diluted sample was dropped and dried on a carbon-coated copper grid. Size distribution was determined by manually measuring the diameter of 1059 particles on the images, using a software custom designed for this purpose (tem_circlefind by András Wacha, MTA TTK, Hungary).

### Atomic Force Microscopy (AFM)

For imaging PBNP complexes, two-fold diluted samples were applied onto poly-L-lysine (PLL)-coated surfaces. PLL-coated substrate surface was prepared by pipetting 100 μL of PLL (0.1% w/v) onto freshly cleaved mica, followed by incubation for 20 min, repeated rinsing with purified water, and drying with a stream of high-purity nitrogen gas. AFM images were collected in noncontact mode with a Cypher S instrument (Asylum Research, Santa Barbara, CA, USA) at 1 Hz line-scanning rate in air, using a silicon cantilever (OMCL AC-160TS, Olympus, Tokyo, Japan) oscillated at its resonance frequency (270–300 MHz). Temperature during the measurements was 25 ± 1˚C. AFM amplitude-contrast images are shown in this paper. The filter used on the images enhances the details of the amplitude contrast images (mud). AFM images were analyzed by using the built-in algorithms of the AFM driver software (Igor Pro, Wave Metrics Inc., Lake Oswego, OR, USA). Particle statistics was done by analyzing a 2 μm × 2 μm height-contrast image with (n = 178) particles. Maximum height values were taken as the height of particles, and rectangularity was calculated as the ratio of the particle

area to the area of a nonrotated inscribing rectangle. The closer a particle is to a rectangle, the closer this value is to unity.

## Animals

In vivo imaging tests of the PBNP nano systems were carried out in NMRI FOXN nu/nu male mice (Janvier, France). Animals had ad libitum access to food and water and were housed under temperature-, humidity-, and light-controlled conditions. All procedures were conducted in accordance with the ARRIVE guidelines and the guidelines set by the European Communities Council Directive (86/609 EEC) and approved by the Animal Care and Use Committee of Semmelweis University (protocol number: PE/EA/1468-8/2019). Mice were 10–12 weeks old with an average body weight of 27 ± 7 g. During imaging, animals were kept under anesthesia using a mixture of 2.5% isoflurane gas and medical oxygen. Their body temperature was maintained at 37°C throughout imaging. For the most humane termination of the animals, intravenous Euthasol (pentobarbital/phenytoin) injection was used.

### *In vitro* and *in vivo* MRI measurements

MRI measurements were performed *in vitro* with a nanoScan® PET/MR system (Mediso, Hungary), having a 1 T permanent magnetic field, 450 mT/m gradient system using a volume transmit/receive coil with a diameter of 60 mm. MRI T1 relaxation rates and r1 relaxivity were calculated from inversion prepared snapshot gradient echo (T1 map, IR GRE SNAP 2D) images acquired with 60 x 90 mm FOV (field of view), plane resolution of 1 mm, slice thickness of 5 mm, 6 averages, TR/TE 4005/1.7, TI 10, 60, 100, 150, 200, 250, 300, 350, 400, 500, 700, 900, 1200, 2500, 4000 ms. MRI-signal enhancement of PBNPs was measured for three different Fe (III) concentrations (13.75 mM, 41.25 mM, and 82.5 mM) in 1.5 mL Eppendorf tubes. After scanning, the concentration dependent signal changes were calculated and compared to the signal of saline.

Experiments were performed in an adult male mouse under isoflurane anesthesia (5% for induction and 1.5–2% to maintain the appropriate level of anesthesia; Baxter, Arrane). Precisely, 300 μL of IR820-labelled PBNP solution containing 3 mg of Fe (III) in a 30 mg/mL concentration PBNP solution was administered intravenously into the tail. The T1-weighted MRI biodistribution images were collected at two different time points (pre- and post-injection) The MRI scans were performed with gradient echo (T1 GRE 3D) images acquired with 100 mm x 40 mm FOV, matrix size 200 x 80, slice thickness of 0.5 mm, 4 averages, TR/TE 75/4, dwell time 25 ms. Images were further analyzed with Fusion (Mediso Ltd., Hungary) and VivoQuant (inviCRO LLC, US) dedicated image analysis software.

### *In vitro* and *in vivo* Fluorescence-labeled Organism Bioimaging Instrument (FOBI) measurements

The fluorescent labelled PBNPs were imaged using a two-dimensional epifluorescent optical imaging instrument. (FOBI, Neoscience Co. Ltd., Suwon-si, Korea). For *in vitro* scans, 0.5 mL of samples were tested with the following imaging parameters: excitation at 680 nm corresponding to the excitation maximum of the dye (excitation: 690 nm; emission: 820 nm), exposure time: 1000 msec and gain: 1. The emission spectrum of the dye was in the pass band of the used emission filter.

Experiments were performed in an adult male mouse under isoflurane anesthesia (5% for induction and 1.5–2% to maintain the appropriate level of anesthesia; Baxter, Arrane). Precisely, 300 μL of IR820-labelled PBNP solution was administered intravenously into the tail vein. The biodistribution images were collected at two different time points (pre- and post-

injection) with excitation of 680 nm corresponding to the excitation maximum of the dye (excitation: 690 nm; emission: 820 nm). The emission spectrum of the dye was in the pass band of the used emission filter. Image acquisition parameters were the following: exposure time: 1000 msec and gain: 1. The images were evaluated with VivoQuant software (Invicro, 27 Drydock Avenue, Boston, MA, USA).

## Results and discussion

### The structure of PBNPs

The final PBNP complex nano structure was prepared by the combination of Shokouhimehr's method [36] and our previously published one-step citrate coated PBNP procedure [22]. The synthesis of PBNPs with and without coating resulted in two different types of PBNP solutions, which were mixed. During the incubation and mixing period, the particles were able to connect to each other via carboxyl groups of citric acid and a form a bigger and iron-richer formula. The porous surface of the nanoparticles assured the conjugation points for the fluorescent IR820 dye (Fig 1).

### Output parameters of nanoparticle characterization methods

Many articles investigated the differences between the possible methods used for characterization of nano-sized objects, nanosuspensions and nanoparticles. Even though in the fields of

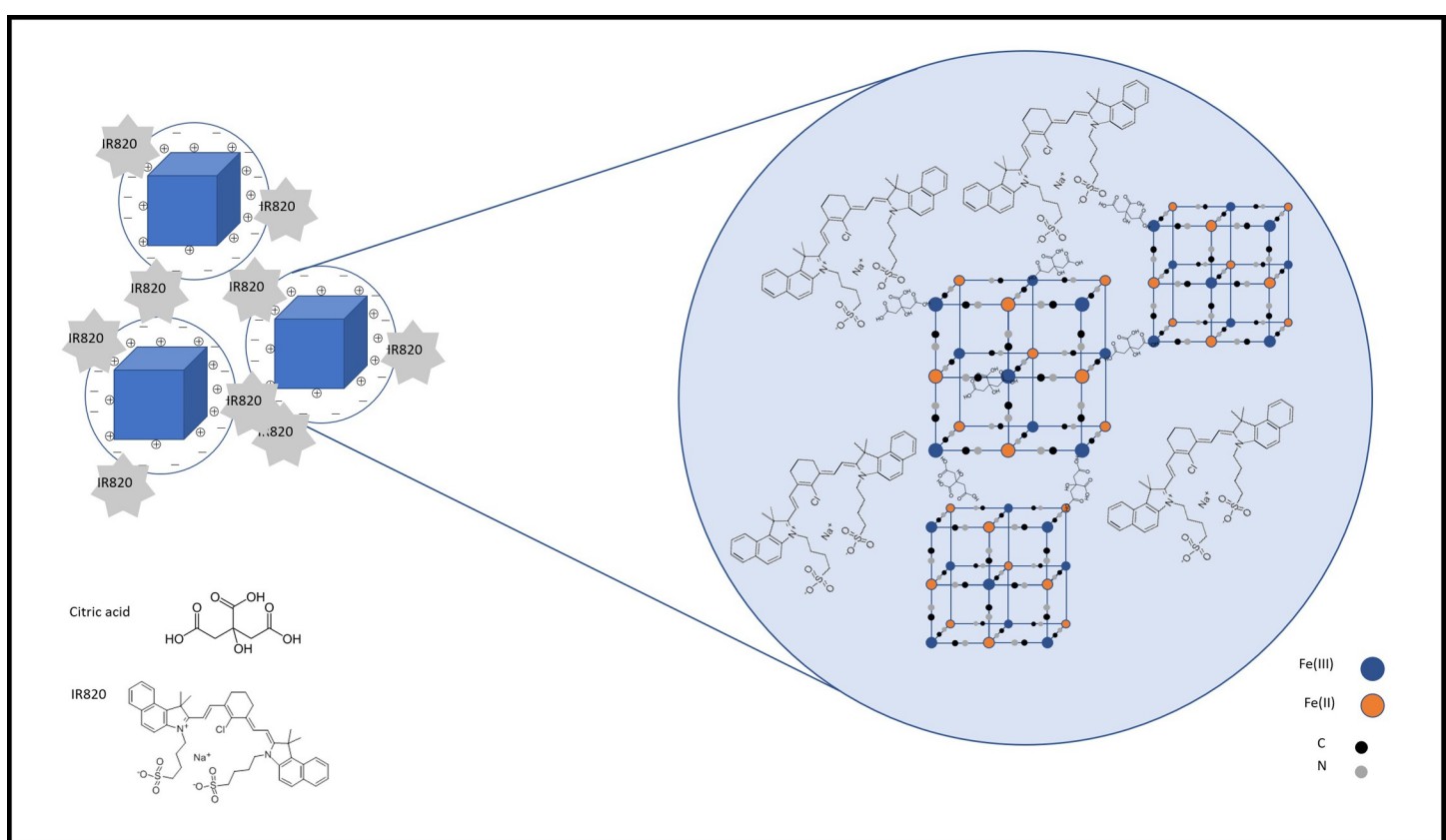

**Fig 1. Schematic illustration of the presumed connection of Prussian blue nanoparticles (PBNPs) with and without coating and the particle conjugation by IR820.** The blue halo around the PBNPs represent the non-biocompatible PBNP-species (PB-HCl) acting as a coating surface. The colors represent the following ions or atoms, respectively: blue: Fe (III); orange: Fe (II); black: C; gray: N.

materials science and chemical engineering, there is a strong need for different types of measurements of the same materials, however the interpretation and the proper understanding of each method is needed to achieve the desired goals. The most frequently used methods to describe a nano system are the DLS, the AFM and the TEM. These methods differ from each other regarding the mathematical basics, the methods of, sensitivity and robustness; a direct comparison is unattainable, hence in most studies, not only one size-range, but a size distribution in form of either a histogram or figure is found [38–40].

For a better understanding of our results, the raw measurement data to our article in the supplemental information section of the manuscript is attached (S1-S4 Figs in S1 File).

## DLS and Zeta potential

The applied citric acid as surface-capping agent controlled the size and the biocompatibility of the synthetized particles and seemed an appropriate agent to avoid agglomeration [41]. The created nanoparticles were a colloidally stable system. The mean hydrodynamic diameter (intensity-based harmonic average) of complex PBNPs was 82.91 ± 1.21 (average ± SD), as determined by DLS. This had only changed slightly with time. There was no significant colloidal alteration during the 4-week duration of the study, as the calculated 0.244 ± 0.014 polydispersity index (PDI) shows the PBNPs did not flocculate or aggregate during this time (not illustrated). The mean zeta potential of PBNPs at the measured pH range did not exceed 15 mV (n = 3). At pH 7.4 the zeta potential was −33.3 ± 3.8 mV (n = 3).

## Atomic force microscopy

AFM is a widely used imaging modality to measure and manipulate sub-nanometer samples [39]. During a measurement only the height of the particles could be determined due to the tip convolution which leads to artificially modified lateral dimensions on the images [42]. The measured width of the particles was influenced by tip convolution. Fig 2 shows PBNPs on AFM images as objects with a flat rectangular surface protruding from a rounded halo. The rectangular surface represents the real geometry of the particles while their halo is the consequence of tip convolution, i.e., the effect of imaging a rectangular prism by a tetrahedral AFM tip. Rectangularity of the particles (together with their halo) was found to be 0.774 ± 0.111 (mean ± SD), indicating that PBNPs indeed represent rectangular topography. The height of the particles was 36.457 ± 9.496 nm (mean ± SD) (Fig 2).

## Transmission electron microscopy

The non-hydrated shape and size of the PBNPs were investigated with TEM. PBNPs appeared flat rectangular, dense objects in this case as well. The mean diameter of the nanoparticles was 30.14 ± 10.656 nm (average ± SD) (Fig 3), along with an average surface area of 579.257 ± 398.983 nm$^2$ (mean ± SD; n = 1059 particles). The measured height by TEM was in good correlation with the results of AFM measurements describing the shape of non-hydrated particles. By both cases, the flat rectangular objects represent the real geometry of the particles (Fig 3).

## Magnetic resonance imaging

To demonstrate the positive MR contrast enhancing property of our PBNP sample, T1-weighted images of a phantom (containing three different Fe (III) concentrations (13.75 mM, 41.25 mM, and 82.5 mM) containing PBNP solutions) were scanned to visually evaluate the signal enhancement on T1-weighted image. Based on the inversion prepared gradient echo

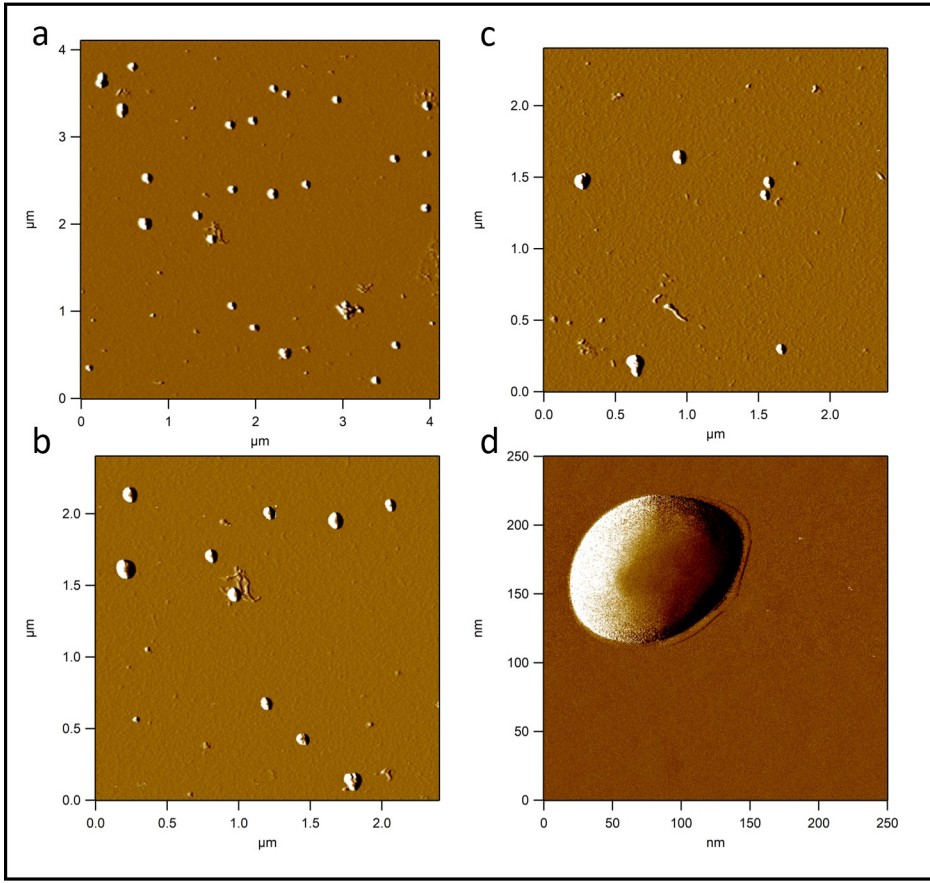

**Fig 2. Atomic force microscopy (AFM) amplitude-contrast images with different magnification of PBNPs on mica surface.** PBNPs on AFM images as objects with a flat rectangular surface protruding from a rounded halo. The rectangular surface represents the real geometry of the particles while their halo is the consequence of tip convolution. Rectangularity of the particles (together with their halo) was found to be 0.774 ± 0.111 (mean ± SD), indicating that PBNPs indeed represent rectangular topography. The height of the particles was 36.457 ± 9.496 nm (mean ± SD). The size of the images are 4.125 μm x 4.125 μm (Fig 2A), 2.5 μm x 2.5 μm (Fig 2B and 2C) and 250 nm x 250 nm (Fig 2D) respectively.

scan and the multislice multiecho scan T1 relaxations rate were calculated. Afterward from these values, longitudinal relaxivity (1 = 0.0008 ± 0.0002 mM−1 ms−1) was calculated. The more significant T1 shortening effect for PBNPs could be explained by a carbon-bound and low-spin of $Fe^{2+}$ in the PB structure, in contrast to the high spin nitrogen-bound $Fe^{3+}$ [36]. Our result demonstrates that PBNPs have substantial T1 MRI contrast compared to other T1 CAs [43, 44].

Nanoparticles without any conjugated specific *in vivo* targeting agent are initially dispersed in the circulation system and started to accumulate mainly in the reticuloendothelial system (RES; e.g. liver, spleen) [45, 46]. To investigate the PBNP uptake efficiency, especially in RES, the PBNP distribution was determined on T1-weighted MR images (Fig 4). In the case of *in vivo* MRI scans, we were able to register contrast changes between the pre- and post-injection scans immediately after the PBNP administration. Enhanced signal intensities were registered in the lungs, liver, kidneys, and abdominal vein (Fig 4), which supports the results of previous publication [47].

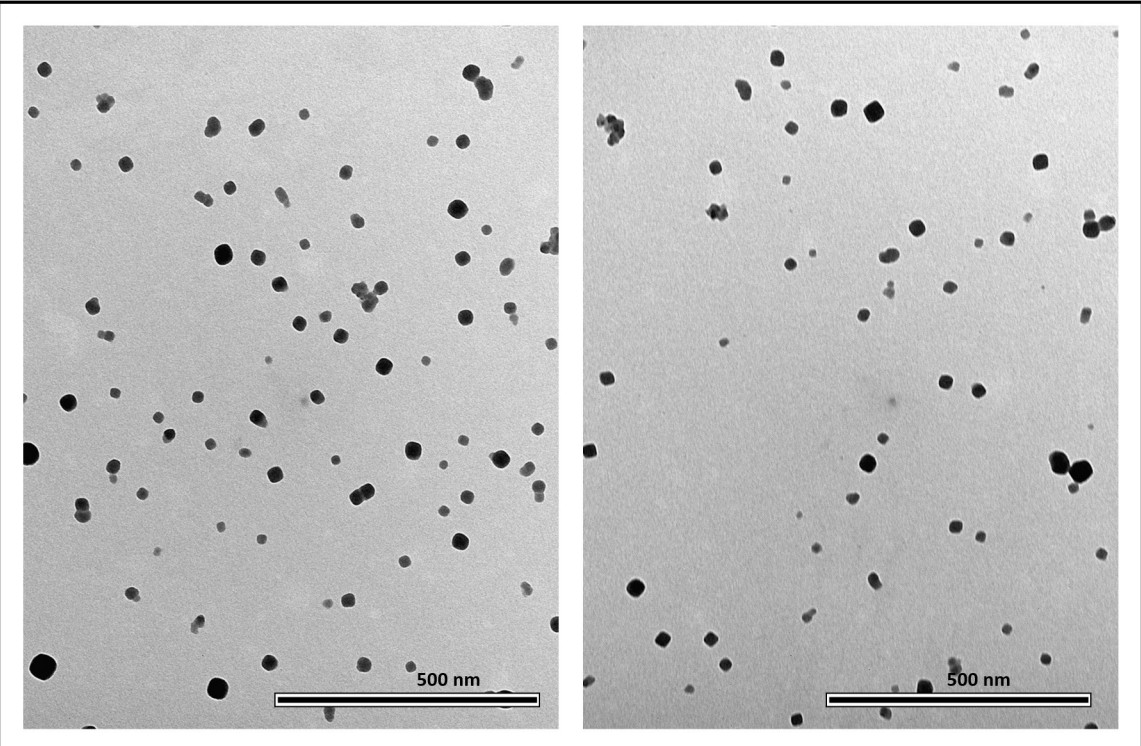

**Fig 3. TEM images of PBNPs on carbon-coated copper grid.** PBNPs appeared flat rectangular, dense objects. The mean diameter of the nanoparticles was 30.14 ± 10.656 nm (average ± SD). The smaller objects on the image are individual PBNP particles which are not conjugated into the final PBNP complex. Scale bar is 500 nm.

## Optical imaging

Due to the strong visible fluorescent signal of IR820 conjugated PBNPs the semiquantitative distribution of the particles was determined based on their normalized mean fluorescent intensity. Fig 5A illustrates the autofluorescence signal from the animal at pre-injection condition at 690 nm, while Fig 5B shows the fluorescent signal after the PBNP administration. Enhanced dye concentration was registered in the head and thoracic region based on the high dye content of the circulation system, furthermore the images illustrated the liver of the animal. According to the studies of Zhang et. al. and Huang et. al., IR820 connected to different types of carrier systems shows great photo- and pH stability, as well as in aqueous media [48, 49] (Fig 5).

## The toxicity of PBNPs

To evaluate the biocompatibility, PBNPs were widely investigated and involved in cellular uptake, cell viability and toxicity studies. Shokouhimehr et. al. reported no possible toxicity of modified PBNPs on HEK-293 cells, furthermore, the cell viability was measured to be ca. 98% [36]. Additionally, Feng and colleagues were modifying PBNPs to make a new type of anticancer drug. Their experiments included the measurement of PBNPs on 4T1 cell line. The reported cell viability in this case was also above 90%, moreover, the relatively high (0.5 mg/mL) PBNP concentrations were also unable to induce cytotoxicity [50].

The cellular uptake of PBNPs was also examined several times; mesenchymal C3HT10T1/2 stem cells (MSC) were treated with the nanoparticles and the results were evaluated using

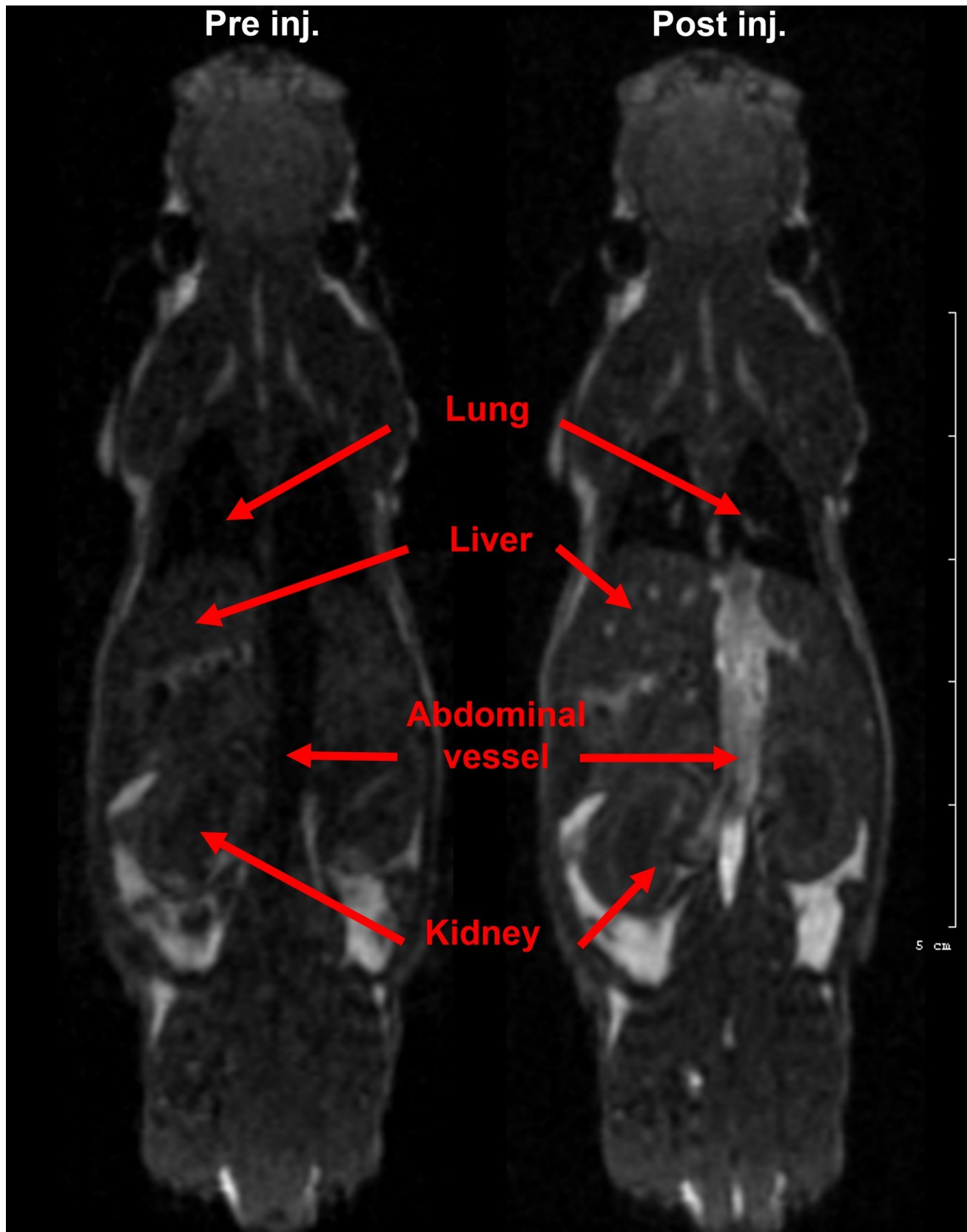

**Fig 4.** Axial T1-weighted MR images of a mouse (A) before and (B) after intravenous administration of Prussian Blue. Red arrows indicate that regions where signal intensity changes happened. Enhanced signal intensities were registered in the lungs, liver, kidneys, and abdominal vessel, which supports the results of a previous publication [47].

TEM. After incubation, PBNPs were detected in the cytoplasm of the MSCs, but the cellular uptake was not yet described. The suggested mechanism of action could be like other types of noble metal and inorganic nanoparticles, which can be taken up via endocytosis, according to Kim et. al., Lu et. al. and Pan et. al. [51–54]. The cytotoxicity on the MSCs was studied for 72 hours, yet these experiments also suggested the lack of toxic effect as well as the lack of influence on the proliferation of MSCs [51].

Based on the broad spectrum of data regarding this matter, we conclude, that our PBNP-complexes have no potential cytotoxic nor proliferation-influencing effect on the cells of living organisms. Regarding the in vivo toxicity, our previous measurements suggested that PBNPs would be excreted by biliary as well as renal routes, mostly during a 72-hour period [22, 55].

## Conclusions

In conclusion, the synthesized NIR-820 conjugated PBNP nanoparticles seem an appropriate MRI and optical contrast material. The surface modification of citrate coated PBNPs with coatless nanoparticles produced slightly enlarged, iron rich complex nano system with enhanced *in vitro* and *in vivo* T1-weighted MR contrast. The further conjugation with NIR-820 dye resulted an optically active complex nano material for *in vivo* use. This nano system reported here exhibited high colloidal stability and monodispersity after each modification step. Its relaxivity constants demonstrated that this nano material is an appropriate candidate for further MRI and OI investigation.

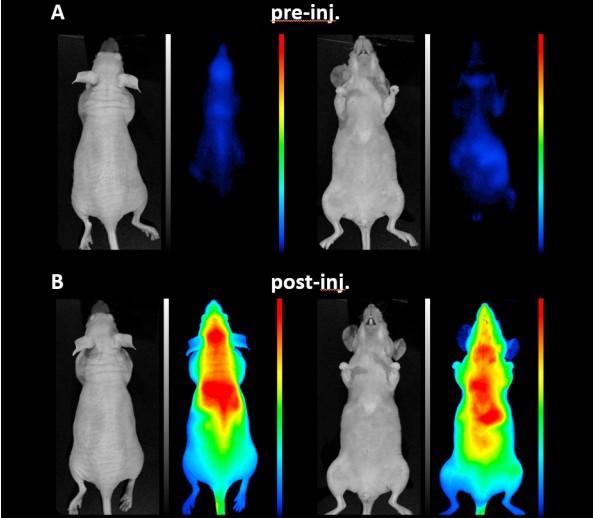

**Fig 5.** White and fluorescent images of a mouse (A) before and (B) after intravenous administration of Prussian Blue (images from left to right: prone white, prone fluorescent, supine white, supine fluorescent images) at 690 nm. After the PBNP administration enhanced dye concentration was registered in the head and thoracic region based on the high dye content of the circulation system, furthermore the images illustrated the liver of the animal. The images are highlighted on the same dynamic color look-up table, which illustrates the signal intensity with different tone from cold to hot colors.

## Supporting information

**S1 File.**
(DOCX)

## Acknowledgments

P. P. and B. G. acknowledges the support from Imaging Probe Development Platform (IPDP) of Lee Kong Chian School of Medicine and from the Cognitive Neuroimaging Centre (CONIC) at Nanyang Technological University, Singapore.

## Author Contributions

**Conceptualization:** Nikolett Hegedűs.

**Methodology:** Nikolett Hegedűs, László Forgách, Bálint Kiss, Zoltán Varga, Bálint Jezsó, Noémi Kovács, Polett Hajdrik, Domokos Máthé.

**Project administration:** Nikolett Hegedűs, Domokos Máthé.

**Supervision:** Balázs Gulyás, Krisztián Szigeti, Domokos Máthé.

**Visualization:** Nikolett Hegedűs, László Forgách, Ildikó Horváth, Noémi Kovács.

**Writing – original draft:** Nikolett Hegedűs.

**Writing – review & editing:** László Forgách, Parasuraman Padmanabhan, Balázs Gulyás, Domokos Máthé.

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
