## [Decision Letter · Decision Letter 0]

4 Oct 2021

PONE-D-21-26225Synthesis and preclinical application of a Prussian blue-based dual fluorescent and magnetic contrast agent (CA)PLOS ONE

Dear Dr. Forgách,

Thank you for submitting your manuscript to PLOS ONE. After careful consideration, we feel that it has merit but does not fully meet PLOS ONE’s publication criteria as it currently stands. Therefore, we invite you to submit a revised version of the manuscript that addresses the points raised during the review process.

There is a need to explain the difference of the present studies from the previous studies of the research group. 

We look forward to receiving your revised manuscript.

Kind regards,

Kaisar Raza

Academic Editor

PLOS ONE

Journal Requirements:

Reviewers' comments:

Reviewer's Responses to Questions

**Comments to the Author**

1. Is the manuscript technically sound, and do the data support the conclusions?

Reviewer #1: Partly

Reviewer #2: Partly

2. Has the statistical analysis been performed appropriately and rigorously? 

Reviewer #1: No

Reviewer #2: N/A

3. Have the authors made all data underlying the findings in their manuscript fully available?

Reviewer #1: No

Reviewer #2: Yes

4. Is the manuscript presented in an intelligible fashion and written in standard English?

Reviewer #1: Yes

Reviewer #2: No

5. Review Comments to the Author

Reviewer #1: Synthesis and preclinical application of a Prussian blue-based dual fluorescent and magnetic contrast agent (CA), I appreciate the authors for the contribution for this manuscript.

However, the manuscript needs major revision in the below mentioned sections.

Abstract: Authors need to mention the brief summary of results in few lines

Authors published one recent article on “Fluorescent, Prussian Blue-Based Biocompatible Nanoparticle System for Multimodal Imaging Contrast” What is the difference and novelty of the current manuscript as compared to published one. Method for the preparation is more or less similar except temperature.

Why there is an elevated temperature used in current method like 60°C as compared to published method 40°C?

In results authors should mention the DLS graphs for representation of particle size distribution

Discussion of the results is not adequate; Authors should elaborate the discussion section in the manuscript

Figure:1 Clarity is missing, authors should change the figure

Figure 2 b: By looking at the TEM image the size of the particles are less than 100 nm which is 49 nm, but the authors mentioned size of complex is 120.41 ± 14.99 nm, kindly provide the TEM and DLS image for the complex as well.

Authors might have provided the toxicity studies of the synthesized complex/CA : in cell line and animal model

Minor corrections

Grammatical errors need to be resolved in the manuscript

Like surface area : nm2, super and subscripts as well as ./, in the manuscript.

Reviewer #2: The manuscript of Hegedus and co-authors describes the “Synthesis and preclinical application of a Prussian blue-based dual fluorescent and magnetic contrast agent (CA)”. The research field is very attractive, but overall the work has nicely curated the approach for the synthesis is not new. My major concern is authors have already published another similar article in Journal Nanomaterials (MDPI) entitled "Fluorescent, Prussian Blue-Based Biocompatible Nanoparticle System for Multimodal Imaging Contrast" (Nanomaterials 2020, 10, 1732). This somewhat dampens the element of novelty to the current work. Thus, I do not believe this work's impact, novelty, and significance warrants publication, and I cannot recommend this MS for publication in Plos One. This Nano-work may be submitted elsewhere in a more appropriate specialized Journal. Apparently, the article can be improved by the authors. Several comments, questions, and recommendations to the authors are provided below:

1. Discuss the element of novelty discussed in the current paper compared to the published article mentioned above (Nanomaterials 2020, 10, 1732).

2. The figures supplied in the manuscript are quite hazy that need to be replaced with better resolution images.

3. What is the fate of these developed nanoparticles upon metabolism?

4. What is the exact duration of detectable fluorescence of these molecules in the cardiovascular system, as the author claims their highest localization in the cardiovascular system.

5. How author thinks the currently developed nanoparticles are efficient in contrast to other approved MRI contrast agents.

6. PLOS authors have the option to publish the peer review history of their article (what does this mean?). If published, this will include your full peer review and any attached files.

Reviewer #1: **Yes: **Nagavendra Kommineni

Reviewer #2: No

---

## [Author Response · Author response to Decision Letter 0]

17 Dec 2021

Dear Dr. Nagavendra Kommineni,

Hereby you may read our answers for the raised questions, suggested changes, and comments. We are thankful for the supportive questions and comments which helped to improve the quality of our manuscript. 

The questions raised by the Reviewers can be read with bold and italic font, the responses are normal case letters. 

Reviewer #1: Synthesis and preclinical application of a Prussian blue-based dual fluorescent and magnetic contrast agent (CA), I appreciate the authors for the contribution for this manuscript.

However, the manuscript needs major revision in the below mentioned sections.

Abstract: Authors need to mention the brief summary of results in few lines

As the reviewer suggested, we added a summary and outlook to the abstract

Authors published one recent article on “Fluorescent, Prussian Blue-Based Biocompatible Nanoparticle System for Multimodal Imaging Contrast” What is the difference and novelty of the current manuscript as compared to published one. Method for the preparation is more or less similar except temperature.

We are more than pleased that the Referee is mentioning our previous work, moreover we are also aware of the fact, he raised questions about it, in context with the novelty regarding this work. However, we strongly reject questioning the novelty of our work. The scope of PlosONE does not cover the innovation potential of a paper; the aim is to distribute original research reports to empower researchers through inclusivity, choice, credit, and transparency. 

Briefly, outcome of the previous work was that Prussian Blue nanoparticles are capable of being labelled with a fluorescent Phenothiazine-like structure (Methylene blue). Since the most commonly used fluorescent dyes have extended aromatic structures (like Indocyanine or Phenothiazine), this step was necessary for further investigation. In addition to this, we executed in vivo fluorescence imaging using this labelled complex, also as a necessary step to lay the foundations for the multimodal imaging

This manuscript takes this whole method and results a step further. Our group has already published an article regarding the SPECT and MRI capabilities of PBNPs (article attached), however we were not combining all the PBNP species which can be fabricated via coprecipitation-synthesis: PBNPs can be precipitated with or without the presence of an organic acid (citric acid); the medium should be however acidic, using hydrochloric acid in the second case.

The new method we show in this work is using a combination biocompatible and non-biocompatible (therefore in an acidic medium made by using hydrochloric acid) PBNP species. We are describing this species as well as the synthesis method regarding this nanoparticles species, which was necessarily modified in order to successfully synthesize the new species – we found an elevated temperature of the synthesis is much more promising for success. What we suspect behind this whole method is a core-shell-structure PBNP species, which has a biocompatible core (PBNP-AC) and one or more layers non-biocompatible PB species (PBNP-HCl). These particles’ contrast enhancing capabilities are orders of magnitude higher than the previous PBNP species we described (they are now in a comparable range with the Gd-based agents); therefore, the novelty of this work is beyond debate. 

To sum up, we show a whole new method for labelling the PBNPs using different type of fluorescent dye (indocyanine-derivate IR820) and showed a novel approach for enhancing the T1 MRI contrast of the PBNP-species making as a potential competitor of the Gd-based MRI contrast agents in the near future. 

 

Why there is an elevated temperature used in current method like 60°C as compared to published method 40°C?

The new method we show with this work is using a combination biocompatible and non-biocompatible (hydrochloric acid used instead of citric acid) PBNP species. We are describing this species as well as the synthesis method regarding this nanoparticles species, which was necessarily modified in order to successfully synthesize the new species – we found an elevated temperature of the synthesis is much more promising for success. 

What we suspect behind this whole method is a core-shell-structure PBNP species, which has a biocompatible core (PBNP-AC) and one or more layers non-biocompatible PB species (PBNP-HCl). These particles’ contrast enhancing capabilities are orders of magnitude higher than the previous PBNP species we described (they are now in a comparable range with the Gd-based agents); therefore, the novelty of this work is beyond debate. 

In results authors should mention the DLS graphs for representation of particle size distribution

The results for the DLS measurements were added as a graph to supplemental information section. 

Discussion of the results is not adequate; Authors should elaborate the discussion section in the manuscript

According to the suggestions of Reviewer 1 and Reviewer 2, we have investigated the possible mechanism of action between PBNP-AC and PBNP-HCl, we added the results to the discussion section.

Figure:1 Clarity is missing, authors should change the figure

Figure 1 was corrected and changed as suggested. 

Figure 2 b: By looking at the TEM image the size of the particles are less than 100 nm which is 49 nm, but the authors mentioned size of complex is 120.41 ± 14.99 nm, kindly provide the TEM and DLS image for the complex as well.

As it is known, DLS measurements have great predictive power regarding the size-distribution of an either monodisperse (polydispersity index ~ 0,1) sample, however it has limited effect on samples having higher PDIs. The 120 nm size of the particles was mentioned in connection with the DLS measurements – clearly, this phenomenon should be elaborated in the discussion section [1-3]

Nevertheless, we executed repeated TEM, AFM and DLS measurements, the new results were included in the recent version of our manuscript. The graph for the DLS measurements was included in the supporting information section. Based on extensive AFM and TEM measurements, the data regarding the size and shape distribution was added to the supplemental information section. We also investigated the possible methods to enhance the monodispersity of the sample; the DLS measurement data is available in the supplemental information section. 

Authors might have provided the toxicity studies of the synthesized complex/CA : in cell line and animal model

We appreciate the Reviewer’s suggestion regarding toxicity data; a sub-paragraph was added to the Results and Discussion session, where several articles were cited regarding this matter [4-9]. 

Minor corrections

Grammatical errors need to be resolved in the manuscript

We were correcting the grammatical shortcomings of the manuscript, according to the suggestion of the Reviewer. 

Like surface area : nm2, super and subscripts as well as ./, in the manuscript.

The suggested corrections were made accordingly, the size-distribution of the particles based on the 3 characterization methods are available in the supplemental information section of the manuscript. 

Reviewer #2: The manuscript of Hegedus and co-authors describes the “Synthesis and preclinical application of a Prussian blue-based dual fluorescent and magnetic contrast agent (CA)”. The research field is very attractive, but overall the work has nicely curated the approach for the synthesis is not new. 

My major concern is authors have already published another similar article in Journal Nanomaterials (MDPI) entitled "Fluorescent, Prussian Blue-Based Biocompatible Nanoparticle System for Multimodal Imaging Contrast" (Nanomaterials 2020, 10, 1732). 

This somewhat dampens the element of novelty to the current work. Thus, I do not believe this work's impact, novelty, and significance warrants publication, and I cannot recommend this MS for publication in Plos One. 

This Nano-work may be submitted elsewhere in a more appropriate specialized Journal. Apparently, the article can be improved by the authors. Several comments, questions, and recommendations to the authors are provided below:

1. Discuss the element of novelty discussed in the current paper compared to the published article mentioned above (Nanomaterials 2020, 10, 1732).

We are more than pleased that the Referee is mentioning our previous work, moreover we are also aware of the fact, he raised questions about it, in context with the novelty regarding this work. However, we strongly reject questioning the novelty of our work. The scope of PlosONE does not cover the innovation potential of a paper; the aim is to distribute original research reports to empower researchers through inclusivity, choice, credit, and transparency. 

Briefly, our previous work was that Prussian Blue nanoparticles are capable of being labelled with a fluorescent Phenothiazine-like structure (Methylene blue). Since the most commonly used fluorescent dyes have extended aromatic structures (like Indocyanine or Phenothiazine), this step was necessary for further investigation. In addition to this, we executed in vivo fluorescence imaging using this labelled complex, also as a necessary step to lay the foundations for the multimodal imaging

This manuscript takes this whole method and results a step further. Our group has already published an article regarding the SPECT and MRI capabilities of PBNPs (article attached), however we were not combining all the PBNP species which can be fabricated via coprecipitation-synthesis: PBNPs can be precipitated with or without the presence of an organic acid (citric acid); the medium should be however acidic, using hydrochloric acid in the second case.

The new method we show in this work is using a combination biocompatible and non-biocompatible (therefore in an acidic medium made by using hydrochloric acid) PBNP species. We are describing this species as well as the synthesis method regarding this nanoparticles species, which was necessarily modified in order to successfully synthesize the new species – we found an elevated temperature of the synthesis is much more promising for success. What we suspect behind this whole method is a core-shell-structure PBNP species, which has a biocompatible core (PBNP-AC) and one or more layers non-biocompatible PB species (PBNP-HCl). These particles’ contrast enhancing capabilities are orders of magnitude higher than the previous PBNP species we described (they are now in a comparable range with the Gd-based agents), therefore the novelty of this work is beyond debate. 

To sum up, we show a whole new method for labelling the PBNPs using different type of fluorescent dye (indocyanine-derivate IR820) and showed a novel approach for enhancing the T1 MRI contrast of the PBNP-species making as a potential competitor of the Gd-based MRI contrast agents in the near future. 

2. The figures supplied in the manuscript are quite hazy that need to be replaced with better resolution images.

We would like to thank the Referee for informing us about the shortcomings of our manuscript. Even though we tried to ensure the best quality pictures, mistakes are inevitable. We tried to improve the resolution; according to the requirements of PlosONE they are suitable for publication. A possible reason behind the Referees’ experiences regarding the poor image quality that might be due to the PDF compression and program could compress the images redundantly. The separately uploaded images were double checked and corrected as requested. 

3. What is the fate of these developed nanoparticles upon metabolism?

According to the published data, several elimination routes of PBNPs are already known. These references are included in the appropriate section of the manuscript. [4-9]

4. What is the exact duration of detectable fluorescence of these molecules in the cardiovascular system, as the author claims their highest localization in the cardiovascular system.

FOBI device is rather a semi-quantitative method for measuring fluorescence in living organisms therefor it cannot provide quantitative data for fluorescence (concentration or fluorescent lifetime). Furthermore, our fluorescent measurements that have lasted about 20 minutes, we did not experience any signal decrease in vivo.

Notwithstanding, we cited several articles regarding the fluorescent lifetime and degradation of IR820 in vivo. [10,11]

5. How author thinks the currently developed nanoparticles are efficient in contrast to other approved MRI contrast agents.

Our recently introduced material is not yet in clinical phase of testing, what we would like to emphasize in this manuscript is that PBNPs could be potential candidates to replace the clinically used materials containing Gd. 

As per latest decision of the European Medicines Agency (EMA) dated 2017 December 17th, contrast agents containing Gd and linear chelators were suspended and withdrawn from the market starting June 2018, due to their potential toxicity on the CNS (EMA/625317/2017). That leaves physicians with the only choice to use Gd-CAs chelated with macrocyclic molecules and still leaves the question unanswered whether the Gd-containing materials can be trusted in the clinical practice. Approaching the problem of such materials could be the first step for a revolutionary change in the clinical practice towards innovative, safe, and efficient medicines and medical devices. 

According to the annual financial report of one of the biggest MRI CA manufacturers, Guerbet Group, their sales have dropped about 15% in the year 2020 in the segment of CAs, which indicates a continuously decreasing tendency in this field in the past years [12]. 

As a conclusion, there is an urge for the development of the new generation MRI CAs, which can be only initiated by raising the question by publishing our most recent results in this field. What we achieved we brought the PBNPs to a comparable range of T1 relaxation times with the Gd-containing CAs. With this article we are looking forward to increasing the development of PBNP-based systems leading to spread these materials more widely in the clinical trials and clinical practice. 

References: 

1. Tomaszewska, E., Soliwoda, K., Kadziola, K., Tkacz-Szczesna, B., Celichowski, G., Cichomski, M., ... & Grobelny, J. (2013). Detection limits of DLS and UV-Vis spectroscopy in characterization of polydisperse nanoparticles colloids. Journal of Nanomaterials, 2013.

2. Hoo, C. M., Starostin, N., West, P., & Mecartney, M. L. (2008). A comparison of atomic force microscopy (AFM) and dynamic light scattering (DLS) methods to characterize nanoparticle size distributions. Journal of Nanoparticle Research, 10(1), 89-96.

3. Eaton, P., Quaresma, P., Soares, C., Neves, C., De Almeida, M. P., Pereira, E., & West, P. (2017). A direct comparison of experimental methods to measure dimensions of synthetic nanoparticles. Ultramicroscopy, 182, 179-190.

4. Feng, T., Wan, J., Li, P., Ran, H., Chen, H., Wang, Z., & Zhang, L. (2019). A novel NIR-controlled NO release of sodium nitroprusside-doped Prussian blue nanoparticle for synergistic tumor treatment. Biomaterials, 214, 119213.

5. Wen, J., Zhao, Z., Tong, R., Huang, L., Miao, Y., & Wu, J. (2018). Prussian blue nanoparticle-labeled mesenchymal stem cells: Evaluation of cell viability, proliferation, migration, differentiation, cytoskeleton, and protein expression in vitro. Nanoscale research letters, 13(1), 1-10.

6. Kim, T., Lemaster, J. E., Chen, F., Li, J., & Jokerst, J. V. (2017). Photoacoustic imaging of human mesenchymal stem cells labeled with Prussian blue–poly (l-lysine) nanocomplexes. ACS nano, 11(9), 9022-9032

7. Lu, J., Ma, S., Sun, J., Xia, C., Liu, C., Wang, Z., ... & Gu, Z. (2009). Manganese ferrite nanoparticle micellar nanocomposites as MRI contrast agent for liver imaging. Biomaterials, 30(15), 2919-2928.

8. Pan, D., Caruthers, S. D., Hu, G., Senpan, A., Scott, M. J., Gaffney, P. J., ... & Lanza, G. M. (2008). Ligand-directed nanobialys as theranostic agent for drug delivery and manganese-based magnetic resonance imaging of vascular targets. Journal of the American Chemical Society, 130(29), 9186-9187..

9. Forgách, L., Hegedűs, N., Horváth, I., Kiss, B., Kovács, N., Varga, Z., ... & Máthé, D. (2020). Fluorescent, Prussian Blue-Based Biocompatible Nanoparticle System for Multimodal Imaging Contrast. Nanomaterials, 10(9), 1732.

10. Zhang, D., Zhang, J., Li, Q., Tian, H., Zhang, N., Li, Z., & Luan, Y. (2018). pH-and enzyme-sensitive IR820–paclitaxel conjugate self-assembled nanovehicles for near-infrared fluorescence imaging-guided chemo–photothermal therapy. ACS applied materials & interfaces, 10(36), 30092-30102.

11. Huang, C., Hu, X., Hou, Z., Ji, J., Li, Z., & Luan, Y. (2019). Tailored graphene oxide-doxorubicin nanovehicles via near-infrared

12. Guerbet 2020 annual results, Available from: https://www.guerbet.com/news/2020-annual-results

13.

---

## [Decision Letter · Decision Letter 1]

14 Feb 2022

Synthesis and preclinical application of a Prussian blue-based dual fluorescent and magnetic contrast agent (CA)

PONE-D-21-26225R1

Dear Dr. Forgách,

We’re pleased to inform you that your manuscript has been judged scientifically suitable for publication and will be formally accepted for publication once it meets all outstanding technical requirements.

Kind regards,

Kaisar Raza

Academic Editor

PLOS ONE

Additional Editor Comments (optional):

Reviewers' comments:

Reviewer's Responses to Questions

**Comments to the Author**

1. If the authors have adequately addressed your comments raised in a previous round of review and you feel that this manuscript is now acceptable for publication, you may indicate that here to bypass the “Comments to the Author” section, enter your conflict of interest statement in the “Confidential to Editor” section, and submit your "Accept" recommendation.

Reviewer #1: All comments have been addressed

Reviewer #2: All comments have been addressed

2. Is the manuscript technically sound, and do the data support the conclusions?

Reviewer #1: Yes

Reviewer #2: Partly

3. Has the statistical analysis been performed appropriately and rigorously? 

Reviewer #1: Yes

Reviewer #2: Yes

4. Have the authors made all data underlying the findings in their manuscript fully available?

Reviewer #1: Yes

Reviewer #2: (No Response)

5. Is the manuscript presented in an intelligible fashion and written in standard English?

Reviewer #1: Yes

Reviewer #2: Yes

6. Review Comments to the Author

Reviewer #1: (No Response)

Reviewer #2: I find the authors have made substantial changes to the manuscript in the light of comments. However, small amout of English errors in manuscript are still exist. After English correction, it can be accepted for publication.

7. PLOS authors have the option to publish the peer review history of their article (what does this mean?). If published, this will include your full peer review and any attached files.

Reviewer #1: No

Reviewer #2: No

---

## [Editor Report · Acceptance letter]

17 Mar 2022

PONE-D-21-26225R1 

Synthesis and preclinical application of a Prussian blue-based dual fluorescent and magnetic contrast agent (CA) 

Dear Dr. Forgách:

I'm pleased to inform you that your manuscript has been deemed suitable for publication in PLOS ONE. Congratulations! Your manuscript is now with our production department. 

Kind regards, 

on behalf of

Dr. Kaisar Raza 

Academic Editor

PLOS ONE